# Injectable Carrageenan/Green Graphene Oxide Hydrogel: A Comprehensive Analysis of Mechanical, Rheological, and Biocompatibility Properties

**DOI:** 10.3390/polym16162345

**Published:** 2024-08-19

**Authors:** Danny Moncada, Rebeca Bouza, Maite Rico, Saddys Rodríguez-Llamazares, Natalia Pettinelli, Alana Aragón-Herrera, Sandra Feijóo-Bandín, Oreste Gualillo, Francisca Lago, Yousof Farrag, Horacio Salavagione

**Affiliations:** 1CITENI, Grupo de Polímeros, Campus Industrial de Ferrol, Universidade da Coruña, 15403 Ferrol, Spain; d.moncadav@udc.es (D.M.); maite.rico@udc.es (M.R.); 2Centro de Investigación de Polímeros Avanzados, Edificio Laboratorio CIPA, Av. Collao 1202, Concepción 4051381, Chile; s.rodriguez@cipachile.cl (S.R.-L.); n.pettinelli@cipachile.cl (N.P.); 3IDIS (Instituto de Investigación Sanitaria de Santiago), Cellular and Molecular Cardiology Research Unit, Santiago University Clinical Hospital, Building C, Travesía da Choupana S/N, 15706 Santiago de Compostela, Spain; alannah.aragon@gmail.com (A.A.-H.); sandra.feijoo@gmail.com (S.F.-B.); francisca.lago.paz@sergas.es (F.L.); 4NEIRID Group (Neuroendocrine Interactions in Rheumatology and Inflammatory Diseases), IDIS (Instituto de Investigación Sanitaria de Santiago), Santiago University Clinical Hospital, Building C, Travesía da Choupana S/N, 15706 Santiago de Compostela, Spain; oreste.gualillo@sergas.es (O.G.); yousof.farrag@gmail.com (Y.F.); 5Departamento de Física de Polímeros, Elastómeros y Aplicaciones Energéticas, Instituto de Ciencia y Tecnología de Polímeros (ICTP-CSIC), C/Juan de la Cierva 3, 28006 Madrid, Spain; horacio.salavagione@csic.es

**Keywords:** injectable hydrogels, physical crosslinking, carrageenan, oxidized graphene

## Abstract

In this work, physically crosslinked injectable hydrogels based on carrageenan, locust bean gum, and gelatin, and mechanically nano-reinforced with green graphene oxide (GO), were developed to address the challenge of finding materials with a good balance between injectability and mechanical properties. The effect of GO content on the rheological and mechanical properties, injectability, swelling behavior, and biocompatibility of the nanocomposite hydrogels was studied. The hydrogels’ morphology, assessed by FE-SEM, showed a homogeneous porous architecture separated by thin walls for all the GO loadings investigated. The rheology measurements evidence that G′ > G″ over the whole frequency range, indicating the dominant elastic nature of the hydrogels and the difference between G′ over G″ depends on the GO content. The GO incorporation into the biopolymer network enhanced the mechanical properties (ca. 20%) without appreciable change in the injectability of the nanocomposite hydrogels, demonstrating the success of the approach described in this work. In addition, the injectable hydrogels with GO loadings ≤0.05% *w*/*v* exhibit negligible toxicity for 3T3-L1 fibroblasts. However, it is noted that loadings over 0.25% *w*/*v* may affect the cell proliferation rate. Therefore, the nano-reinforced injectable hybrid hydrogels reported here, developed with a fully sustainable approach, have a promising future as potential materials for use in tissue repair.

## 1. Introduction

Injectable hydrogels have long been a material of study to be used in biomedical and/or pharmaceutical applications thanks to their close resemblance to the extracellular matrix. They can be administrated with minimally invasive procedures that reduce recovery times and risks of infection [1,2]. Their properties can be controlled depending on several factors, e.g., their origin (natural, synthetic or hybrid), method of preparation [3,4,5], and their crosslinking (physical or chemical), which allows them to be used in several applications such as tissue engineering and wound healing [6].

Among a great variety of compounds used for the preparation of hydrogels, there is a natural polymer derived from red algae of the class Rhodophyceae, carrageenans, which are a family of anionic hydrophilic linear sulfated galactan, formed by alternating polymeric units of D-galactose and 3,6-anhydro-D-galactose (3,6-AG) linked through α-1,3 and β-1,4-glycosidic bonds [7,8]. Of the carrageenan family, three main types extracted from different types of algae stand out: kappa carrageenan (κ-C), iota carrageenan (ι-C), and lambda carrageenan (λ-C), which differ in their structure, molecular weight, the amount and number of ester sulfate groups, as well as the content of 3,6-AG that can vary depending on the species of seaweed from which the polymer is obtained [9,10]. These differences bring different properties to each type of carrageenan: solubility, the ability to interact with metal ions, and gelling behavior [11]. κ-C gels are characterized as strong and stiff gels, more brittle in the presence of potassium ions (K^+^). ι-C gels stand out for being more elastic and softer (viscous) gels in the presence of calcium ions (Ca^2+^). Despite that, there is information indicating that it is possible to gel λ-C through trivalent ions (Fe^3+^) [12,13].

The gelation of κ-C and ι-C occurs due to the orientation of the sulfate groups towards the external face of the three-dimensional double helix network structure, thus allowing the crosslinking of the chains. This does not occur in λ-C, where the sulfate groups are oriented towards the internal face of the structure, which prevents crosslinking [14]. Upon dissolution of these polysaccharides at a given temperature range, and subsequent cooling, a coil-to-helix conformational transition occurs, which is then arranged as ordered helices. The final sol-gel-transition step is made possible by the incorporation of cations into the polysaccharide solution.

The main characteristics of these gels are their high-water absorption capacity, low toxicity, thermo-reversible capacity, and modifiable gelling properties [15]. The characteristics of the gels can be modified to improve the properties of the gels depending on the intended application. One possibility is the presence of other components in their structure (xylose, glucose, methyl esters, pyruvate groups, etc.). The other option is by mixing with other compounds, for example, different carrageenans, gelatin, cellulose, locust bean gum, etc. [16,17,18]. 

In the last decade, the use of graphene has been investigated in biomedicine and pharmacology, allowing a new focus of study to open. Graphene is a two-dimensional (2D) material formed from a layer of carbon atoms with sp^2^ bonds that are arranged in a hexagonal lattice linked through strong covalent bonds. There are different types of graphene, which differ in their appearance and the number of functional groups attached to the structure. The most used are graphene oxide (GO) and reduced graphene oxide (rGO) [19,20], which differ in the number of oxygen groups present in their chemical structure, e.g., hydroxyl groups (–OH), carboxylic acids (–COO), carbonyls (C–O), and epoxides [21]. The functional groups present in oxidized or reduced graphene allow interaction with other compounds through covalent and non-covalent bonds. The characteristics of graphene, such as its electrical, mechanical, and thermal properties, allow the properties of gels to be improved [22,23,24]. The development of hybrid injectable hydrogels of carrageenans incorporating graphene in their preparation has been proposed, allowing new research possibilities to open, and thus for novel alternatives that could serve different purposes to be tested. Some examples are the preparation of nanocarriers for controlled and targeted drugs based on GO, grafted with biotin-conjugated κ-C for cancer treatment [25], tissue engineering using κ-C and dopamine-functionalized GO [26], the fabrication of scaffolds for bone regeneration by polymerization of carrageenan, acrylic acid, graphene, hydroxyapatite promoting mineralization, and cell differentiation [27,28].

The possibility of preparing a new injectable biomaterial combining the best properties of each compound would allow its use in different biomedical applications, such as cell transport for wound repair and healing. A disadvantage of using conventional graphene is its high cost and production method. However, there exists an environmentally friendly graphene-manufacturing process capable of producing graphene from biomass, which allows the reduction of production costs and a high-quality graphene powder to be obtained. In this approach, graphene is obtained from biomass through a process that involves the pyrolysis of organic materials, followed by exfoliation to produce high-quality graphene. This method reduces production costs and environmental impact, while yielding graphene with excellent properties [29]. The incorporation of graphene oxide into carrageenan-based hydrogels significantly enhances their mechanical properties, biocompatibility, and injectability, making them highly promising candidates for tissue-repair applications compared to conventional materials such as alginate, polyethylene glycol (PEG), chitosan, hyaluronic acid, and collagen [30,31,32,33].

The objective of this study was the development of a fully sustainable injectable hydrogel through physical crosslinking, using κ-C, ι-C, locust bean gum, gelatin, and oxidized green graphene for their preparation. The prepared injectable hydrogels were characterized by Raman, UV-visible spectroscopy, and TGA. Moreover, the surface morphology, swelling behavior, mechanical properties, and cellular biocompatibility were analyzed.

## 2. Materials and Methods

### 2.1. Materials

The kappa carrageenan (κC) (MW~ 500 kDa), iota carrageenan (ιC) (MW~ 300 kDa), and locust bean gum (LB) (viscosity 3500 cps, 25 °C) were supplied by Ceamsa, Pontevedra, Spain. Gelatin (G) was obtained from Becton, Dickinson and Company, Franklin Lakes, NJ, USA. Potassium chloride and calcium chloride were of analytical grade and purchased from Scharlau, Barcelona, Spain. The hydrophilic green graphene oxide (GO) with approximately 8% oxidation was supplied by GreenTech, Madrid, Spain. This specific oxidation level allows sufficient functional groups (carboxyl, hydroxyl, and epoxide) on the graphene oxide (GO) surface to interact with the hydrogel matrix, enhancing mechanical properties and stability. A moderate level of oxidation ensures that the material remains biocompatible, avoiding cytotoxicity associated with higher oxidation levels. Additionally, 8% oxidation allows for good dispersion within the hydrogel matrix, preventing aggregation and ensuring uniform mechanical reinforcement. All chemicals were used without further purification. The water used in the preparation and the dialysis was purified in a Milli-Q ultrapure system (Millipore, Molsheim, France).

### 2.2. Preparation and Characterization of the Graphene Dispersions

#### 2.2.1. Preparation of Graphene Dispersions

To prepare the graphene dispersions, ιC solutions (1% and 2% *w*/*v*), κC solution (1% *w*/*v*), and gelatin solution (1% *w*/*v*) were made in ultrapure water. These solutions were stirred and heated to 70 °C, 80 °C, and 50 °C, respectively. GO was introduced to each solution at a concentration of 0.5% *w*/*v* and maintained under constant stirring for 15 min. The mixtures were then subjected to ultrasonication for 10 min using a SONOPULS HD 3200 ultrasonic processor (Bandelin electronic GmbH & Co. KG, Berlin, Germany) with a 13 mm diameter titanium microtip. The ultrasonic power applied was up to 200 W at a frequency of 20 kHz. After ultrasonication, the suspensions were centrifuged at 5000 rpm for 3 min. The graphene dispersions were then analyzed using absorbance and Raman spectroscopies. To evaluate the impact of ultrasonication time, the process was repeated with the ultrasonication duration reduced from 10 min to 5 min.

#### 2.2.2. Thermogravimetric Analysis

The thermal stability of green graphene (GO) and its water content were analyzed using a thermogravimetric analyzer (TGA 400, Perkin Elmer, Groningen, The Netherlands). About 15 mg of the graphene was placed in an aluminum container and heated from 50 °C to 800 °C at 5 °C min^−1^ under an inert atmosphere of nitrogen with a flow rate of 20 cm^3^ min^−1^. The procedure was repeated under an oxidizing atmosphere and the results were compared.

#### 2.2.3. Raman Spectroscopy of Graphene Dispersions

Raman spectra were obtained using a Renishaw InVia-Reflex Raman system (Renishaw plc, Wotton-under-Edge, UK). This system features a gratin spectrometer with a Peltier-cooled CCD detector attached to a confocal microscope. An argon ion laser (λ = 785 nm) was used for Raman scattering excitation at 1% power. Each sample was analyzed with 20 accumulated scans using a 100× microscope objective (NA = 0.85). Spectra were collected in the range of 100 to 3200 cm^−1^ from multiple positions. All spectral data were processed using Renishaw WiRE 5.0 software. The analyses included both the graphene powder and the graphene dispersion in a 2% ιC solution. For determining the Raman spectrum of the dispersed sample (ιC+GO), a small droplet was placed on the center of a silica plate for subsequent analysis.

#### 2.2.4. UV-Visible Spectroscopy

To evaluate the dispersion of graphene in different solutions, the absorbances of the diluted graphene dispersions (1/10) were measured in a Lambda 35 UV/VIS Spectrometer, Perkin Elmer. The observed wavelength range was from 400 nm to 800 nm at a rate of 240 nm min^−1^. The results were analysed using uv.winLab v2.1 software. 

### 2.3. Hydrogel Preparation

A stock solution of ιC (2% *w*/*v*), gelatin (1% *w*/*v*), a mixture of κC (1% *w*/*v*) and LB (0.334% *w*/*v*), and the GO suspension in ιC solution were prepared as previously described [18,29]. Graphene was added to the ιC solution (ιC/GO) at concentrations of 0.5% *w*/*v*, 0.25% *w*/*v*, and 0.1% *w*/*v*. The injectable graphene hydrogels (IH+GO) were created by combining 6 mL of κC/LB solution, 1 mL of gelatin solution, and varying amounts of ιC/GO dispersion along with the ιC solution to achieve a final volume of 10 mL. The final graphene concentrations in the hydrogel formulations are detailed in Table 1. The mixtures were stirred magnetically at 55 °C, then 300 µL of 0.5 M KCl and 150 µL of 0.5 M CaCl2 were added to induce gelation. The mixtures were left at room temperature for approximately 1 h to allow complete gelling. A control hydrogel (IH) was prepared by replacing the ιC/GO dispersion with the ιC solution.

### 2.4. Mechanical Properties

The mechanical properties of the hydrogels were assessed using an Instron Universal Testing Machine 5565 (Instron, High Wycombe, UK) in compression mode, equipped with a 100 N load cell and a crosshead speed of 2 mm/min. Cylindrical hydrogel samples with a diameter of 25 mm and a height of 10 mm were utilized for the compression test, which was performed at room temperature using fully swollen hydrogel specimens. The compressive modulus was determined from the slope of the linear region of the stress–strain curves, corresponding to 5–15% strain. The reported values for the compressive modulus and the compressive strain at the break point represent the average of at least six measurements per sample. Statistical analyses of the mechanical properties data were conducted using a one-way analysis of variance (ANOVA) test. The significance between groups for the compressive modulus and the compressive strain at break point was evaluated at *p* < 0.05 using Tukey and Kruskal–Wallis tests, respectively.

### 2.5. Scanning Electron Microscope (SEM)

To evaluate the surface morphological structure, the freeze-dried hydrogels were sputter-coated with iridium using a Cressington 208HR high-resolution sputter coating (Cressington Scientific Instrument, Watford, UK). The surface morphology was observed with a JSM 7200F field emission scanning electron microscope (JEOL USA Inc., Peabody, MA, USA) at an accelerating voltage of 10 kV and 15 kV. The SEM analysis of freeze-dried hydrogel samples provides an understanding of the pore structure, however, it is important to note that the pore sizes observed are not representative of the fully swollen hydrogels in their final application.

### 2.6. X-ray Diffraction

An X-ray diffraction measurement of the injectable hydrogels was carried out using a Bruker Endeavour diffractometer model D4/max-B, (Bruker AXS, Karlsruhe, Germany) with Cu Kα radiation (40 kV and 20 mA). The scanning regions of the diffraction angle (2θ) were 5°–40° in steps of 0.02° and time per step of 0.02° s^−1^. 

### 2.7. Swelling Behavior 

The swelling behaviour measurements were carried out using freeze-dried hydrogels immersed in phosphate buffer saline (PBS) solution at 37 °C. The swollen hydrogels were weighed daily after removing the water until equilibrium or degradation was reached. Surface water of hydrogel was absorbed by filter paper before weighing. The equilibrium swelling ratio (SR) was calculated by Equation (1) [34]. The measurements for all samples were performed in triplicate.
(1)$$SR=\left[\frac{W_{s}-W_{i}}{W_{i}}\right]$$
where W_S_ is the weight of the swollen hydrogel and W*_i_* is the initial weight of the lyophilized hydrogel.

### 2.8. Rheological Properties 

Rheological characterization of the hydrogels was performed using a rheometer Ares (TA Instruments, New Castle, DE, USA) in a parallel plate geometry (diameter 25 mm) cell with a gap between plates of 1 mm. All hydrogel samples were prepared in 12-well plates with a diameter of 25 mm and a height of 2 mm. The elastic modulus (G′) and viscous modulus (G″) were recorded across a frequency range of 0.1–100 rad s^−1^ while maintaining a constant strain of 1%, which falls within the linear range of viscosity. All measurements were duplicated for each sample at a temperature of 25 °C.

To evaluate the smoothness and the injectability, the hydrogels (IH and IH+GO) were led into 10 mL syringes with 18 G needles and injected by hand. The IH was spiked with methylene blue to improve visualization.

### 2.9. Cell Assays

#### Cell Cultures

The American Type Culture Collection (ATTCC) 3T3-L1 fibroblasts, a highly reliable model for preliminary assessments of new biomaterials and extensively used in a wide range of biocompatibility studies, were a gift from Dr. Oreste Gualillo (Institute for health Research, Santiago de Compostela, Spain). 3T3-L1 fibroblasts were cultured in high glucose (4.5 g L^−1^) and Dulbecco’s modified eagle medium (DMEM) (Lonza Group Ltd., Basel, Switzerland), supplemented with 10% heat-in-activated fetal bovine serum (FBS) (Merck KGaA, Darmstadt, Germany), 2% penicillin/streptomycin antibiotics, and 2% L. glutamine (Sigma-Aldrich, St. Louis, MO, USA) at 37 °C and 5% CO_2_.

### 2.10. Biocompatibility Assay

Hydrogels were prepared in a thin monolayer on 35 mm dishes with glass bottoms (Ibidi GmbH, Martinsried, Germany) and left under ultraviolet light for 2 h. 3T3-L1 fibroblasts were seeded at a density of 3 × 10^5^ cell/dish on the hydrogel and cultured for 24 h and 48 h. Cells were simultaneously stained with 5 µM propidium iodide (Merck KGaA, Darmstadt, Germany) and with 2 µM calcein -AM (Thermo Fisher Scientific Inc., Waltham, MA, USA) to identify dead and live cells, respectively. Images were taken using the confocal microscope Leica TCS SP8 MP (Leica Microsystems AG, Wetzlar, Germany).

## 3. Results

The green GO was fully characterized and the results are shown in the electronic Appendix A.

### 3.1. Mechanical Properties 

The effect of GO content on the mechanical properties of the hydrogels was evaluated using a compression test, and the results are presented in Figure 1. The incorporation of GO into the hydrogel matrix increased the compressive modulus (Figure 1a) in comparison with the neat hydrogel for all loadings investigated. When loadings <0.1 *w*/*v*% were employed, an increase in the modulus of ~20% was observed, with this increase being double (40 %) at 0.1 *w*/*v* %. Higher loadings provoked a decrease in the modulus to similar values as those obtained for lower loadings. The improvement of the modulus of the composite hydrogels indicates the formation of effective hydrogel-GO interfacial interactions between the polar functional groups of the GO particles and the biopolymer chains via hydrogen bonding [35], which limits the mobility of the polymer networks and increase the modulus.

Furthermore, the two-dimensional structure of GO can impose a certain degree of geometric constraint on the mobility of polymer chains, which also contributes to modulus enhancement [36]. The observed drop in compressive modulus when increasing GO loading from 0.1% to 0.15% can be attributed to aggregation of GO particles since, at higher concentrations, GO particles tend to aggregate, leading to non-uniform distribution within the hydrogel matrix. This aggregation can create weak points and reduce overall mechanical strength [26]. The modulus values obtained in our GO/carrageenan/gelatin hydrogels are of the same order of magnitude as those reported for similar polysaccharide-based hydrogels reinforced with GO, e.g., alginate-based hydrogel with 0.125 wt.% of GO (~60 kPa) and chitosan/β-glycerophosphate with a GO-loading of 1.0 wt.% [36,37]. Furthermore, our hydrogel demonstrated mechanical strength that was comparable or superior to other commonly used materials for tissue repair, including alginate and hyaluronic acid-based hydrogels [38,39]. Regarding the strain at break, the incorporation of GO does not cause significant changes, regardless of the loading used (Figure 1b). This is interesting since it is expected that the incorporation of stiff GO nanoparticles, effective for increasing the modulus, causes a decrease in the elongation at break. This means that the hydrogels maintain a certain deformability throughout the range of loadings studied, with a strain of >80% without break. 

### 3.2. Scanning Electron Microscopy

The surface morphology of the freeze-dried hydrogels was examined by FE-SEM. The interaction between the hydrogel components (carrageenan/gelatin) generates a porous network with interconnected three-dimensional microstructures (Figure 2), which are important and necessary for vascularization, cellular transport, and nutrient diffusion [18]. Images of the freeze-dried gels showed an irregular surface with porous structure in both pure IH and nanocomposite IH+GO hydrogels. The pore diameter ranges from ca 200 to 550 µm, being larger in the pure hydrogel and decreasing as the GO-loading increases. FE-SEM images suggest that the incorporation of green GO in the formulation of the hydrogels can assist with controlling the pore-size and porosity uniformity [27], forming an interconnected network with morphological characteristics different from the one of the control samples. Furthermore, a change from a smooth surface of the porous network in the pure IH to a rough surface with agglomerations in IH+GO, which increase with GO-loading, is also perceived. 

### 3.3. Rheological Properties

Rheology experiments were carried out to assess the effect of the GO content on the viscoelastic properties of the injectable hydrogels. Figure 3 displays the variation of the storage (G′) and loss (G″) moduli versus angular frequency at a fixed strain of 1% for all samples investigated in this work. For all hydrogels, G′ was higher than G″ in the studied frequency window, indicating the dominant gel-like behavior of these materials under small shear strains. This behavior indicates that the elastic nature of the hydrogels is stronger than their viscous nature [40]. 

G′ of the hydrogels, except IH+GO-0.1 hydrogel, showed a plateau-like behavior between 0.1 and 100 rad/s. In addition, G′ decreased upon increasing GO content, except for hydrogel prepared with 0.15% of GO. Nevertheless, the G′ values ranged from 1660 to 5120 Pa, being comparable with those for soft tissue such as liver, relaxed muscle, and breast glands (10^3^–10^4^ Pa) [41]. In addition, similar values of G′ have been reported for starch/graphene hydrogels [42].

Interestingly, G″ exhibited a non-plateau-like behavior, decreasing at low frequencies and then increasing. This phenomenon can be related to the slight rearrangements in the physical crosslinking network under oscillatory frequency, for which there is no time to rebuild their structure. It has been reported that G″ of physical gels is highly dependent on the oscillatory frequency [43].

Overall, the G′/G″ ratio of the nanocomposites is lower than that for the pristine hydrogel, and decreases as the GO loading increases (Table 2). However, the sample with the highest loading displays a value similar to that of the pure hydrogel (and the nanocomposite with the lowest loading), resembling the results obtained from compression tests. It has been reported that natural tissues present G′/G″ ratio values close to 10 [44], suggesting that our hydrogel with low GO loadings content are potential candidates for tissue engineering applications. 

To examine the flow behavior of the hydrogels, rotational shear tests were conducted. Figure 4a shows that all hydrogels have high apparent viscosities under low shear rates. When the shear rate was increased from 0.01 to 10 s^−1^, the hydrogels showed a pronounced pseudoplastic (shear thinning) behavior and the apparent viscosity decreased by approximately four orders of magnitude. The shear-thinning behavior is characteristic of hydrogels with good injectability [45]. As shown in Figure 4a, all nanocomposite hydrogels, except the one with 0.15 *w*/*v* %, presented slightly lower viscosity than the pristine IH, indicating that low GO concentrations do not restrict the movement of polymer chains in the network. Figure 4b,c visually illustrate that the hydrogel and the nanocomposite hydrogels can be easily injected using a syringe with an 18G-gauge needle and manual pressure.

### 3.4. Swelling Behavior

One of the advantages of most hydrogels based on biopolymers is their water-absorption capability and degradation in physiological conditions. The swelling capacity of the injectable hydrogel can maintain a moist environment and absorb, for example, wound exudate to expedite the healing process. The swelling ratios of hydrogels with and without GO, immersed in PBS solution at 37 °C are shown in the Figure 5. All hydrogels absorbed large amounts of water, reaching on average their maximum absorption between 48 and 72 h, with a swelling ratio of between 50 and 60 (g/g). After 48 h of soaking in PBS, no significant differences were found between the swelling ratio of the hydrogels loaded with different GO contents and the control hydrogel (IH). Results demonstrated that the incorporation of GO in the hydrogel structures did not promote water uptake. The pore size of the hydrogels, according to SEM images, decreases as the graphene content in the hydrogel increases, which may lead to variations in water absorption and pore stability over time [46]. Another aspect to highlight is the stability of hydrogels during the swelling test; the stability of the injectable hydrogels decreased with the increase in GO content. The structure of the IH-GO+0.15% hydrogel broke into fragments after 72 h of immersion in PBS, indicating a higher structural instability of this hydrogel. The decreased stability of the IH with GO could likely be due to a dilution effect of GO in the physical crosslinking carrageenan–gelatin. GO disrupts the biopolymer network in physiological conditions by forming a new unstable network of GO–carrageenan and GO–gelatin, increasing its instability at high concentrations of graphene. Chen et al., 2019 reported that the incorporation of GO into nanocellulose-grafted poly(acrylic acid) composite hydrogels reduced its crosslinking density [47]. The lack of stability of the GO-loaded hydrogels compare to the neat hydrogel is due to a decrease in the crosslinking density.

### 3.5. Biocompatibility Assays 

To evaluate the biocompatibility of our hydrogels, cell adhesion and viability of 3T3-L1 fibroblasts when interacting with the differently prepared injectable hydrogels were analyzed by confocal laser microscopy (CLM). Cell viability is normally assayed by the methylthiazolyldiphenyl-tetrazolium bromide (MTT) test, but CLM was used instead due to the multiple false positive results obtained in the MTT preliminary assays [46]. The cell viability was evaluated at 24 h and 48 h of contact with the hydrogel, after staining with propidium iodide and calcein. Figure 6 shows live cells in green, and dead cells in red. At GO concentrations of ≤0.05% *w*/*v*, the proliferation of 3T3-L1 cells in the hydrogels was like that observed in the control IH [18]. However, when a graphene concentration of 0.075% *w*/*v* was reached, cell proliferation decreased significantly (Figure 6d,h) and their distribution inside the hydrogel was altered, forming agglomerations of cells in different zones. 

The quantitative determination of cell number after 48 h (expressed as cells/mm^2^) and the percentage of live cells are shown in Figure 7. At concentrations higher than 0.025% *w*/*v*, a significant decrease in the 3T3-L1 cell count was observed, compared to the control sample (Figure 7a), indicating a negative interference of GO in the cell proliferation. However, the percentage of live cells/total was similar in all samples (Figure 7b), so no significant effect on the cell viability was noted. Previous studies have shown that the control hydrogels used here exhibit good biocompatibility and proliferation of 3T3-L1 cells [18]. Regarding nanocomposite hydrogels, we have reported that similar injectable hydrogels to those used here, reinforced with pristine graphene (instead of GO), produced materials with improved mechanical performance, swelling, and biocompatibility at loadings ≤0.05% *w*/*v* [46]. 

Considering all the results in the present study, it can be suggested that nanocomposites with GO loadings ≤0.05% *w*/*v* present an optimal balance of properties as well. 

During the last few years, studies on the use of different types of carbon nanomaterials (graphene, GO, rGO, fullerene, or carbon nanotubes) [48] in the field of biomedicine for different applications have remarkably increased, and the results related to their cytotoxicity are diverse [49]. The biocompatibility of the hydrogel is intricately linked to its composition, while the cytotoxicity of graphene-type particles is strongly controlled by the specific cell lines employed and, notably, by the molecular interactions and the concentration of graphene to which they are exposed [46,47]. For instance, various studies have reported different biological responses to the use of graphene in their composition. For instance, it has been shown that the use of nano-GO exhibited positive results in a biocompatibility test with positive results [50]. On the other hand, other studies have described adverse responses when using lung epithelial cells and fibroblast [51]. 

## 4. Conclusions

The preparation of a new injectable hydrogel composed of carrageenan, locust bean gum, gelatin, and green GO, using a fully sustainable approach, was described in this study. The conducted test revealed that, when comparing the pristine hydrogel with hydrogels containing increasing amounts of GO (0.01–0.15%), those with intermediate GO concentrations improved their mechanical strength by up to 15%. These hydrogels maintained high swelling behavior but experienced a decrease in pore size and an increase in roughness as the concentration of graphene in the hydrogels increased. Moreover, the incorporation of GO at these loadings does not affect hydrogel injectability as they preserved the ability to withstand elastic deformation and recover after that. Finally, a cell viability test with 3T3-L1 fibroblast cells showed a slight decrease in cell proliferation as the concentration of GO increased, but the percentage of live cells was maintained. 

The materials developed in this study offer great versatility. The hydrogel system can be tailored for specific tissue-repair applications by adjusting the concentration of GO and the physical crosslinking conditions. Modifying these parameters allows us to optimize the mechanical properties, degradation rate, and biocompatibility for different types of tissue, such as cartilage, bone, or soft tissue.

The in vitro degradation studies demonstrated that the hydrogels retained their structural integrity and mechanical properties over an extended period, indicating good stability. However, further in vivo studies are required to gain a full understanding of the long-term degradation behavior and biocompatibility within a living organism. Further work is being carried out in our laboratory using in vivo animal models to assess tissue-repair capabilities, and the results will be reported soon.

## Figures and Tables

**Figure 1 polymers-16-02345-f001:**
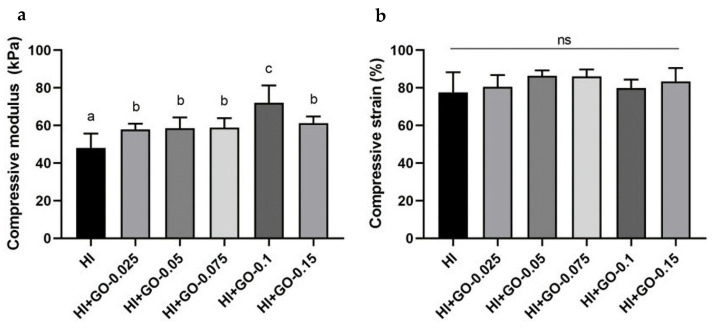
Effect of GO loading on the compressive modulus (**a**) and compressive strain at break point (**b**) of hydrogels. Error bars show standard deviation; *n* = 6. Values with different letters in the columns are statistically different at *p* < 0.05, Kruskal–Wallis test. ns: no statistical difference.

**Figure 2 polymers-16-02345-f002:**
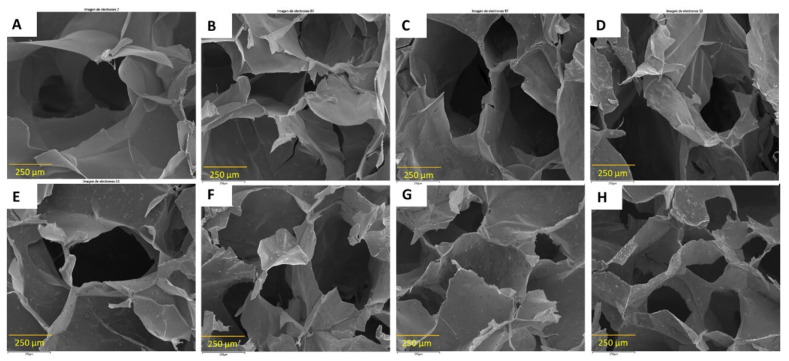
SEM images of (**A**) IH, (**B**) IH+GO-0.01%, (**C**) IH+GO-0.02%, (**D**) IH+GO-0.025%, (**E**) IH+GO-0.05%/, (**F**) IH+GO-0.075%/, (**G**) IH+GO-0.1%/, (**H**) IH+GO-0.15%.

**Figure 3 polymers-16-02345-f003:**
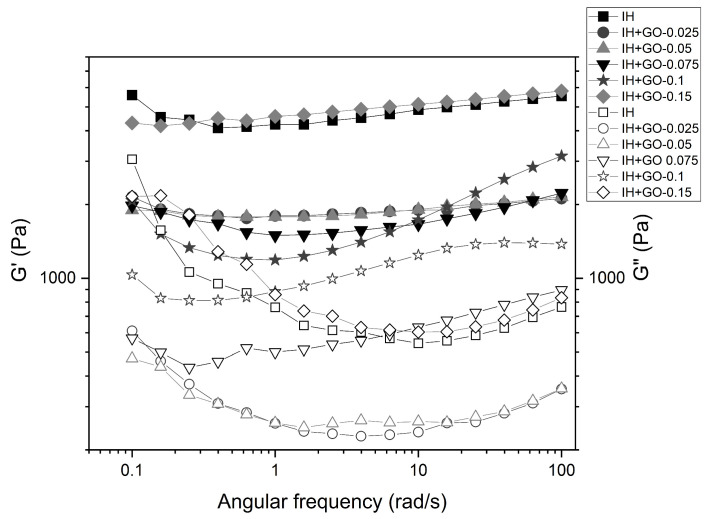
Dependence of the storage modulus (G′, filled symbols) and loss modulus (G″, empty symbols) with the angular frequency at a fixed strain of 1% for HI and HI+GO nanocomposites.

**Figure 4 polymers-16-02345-f004:**
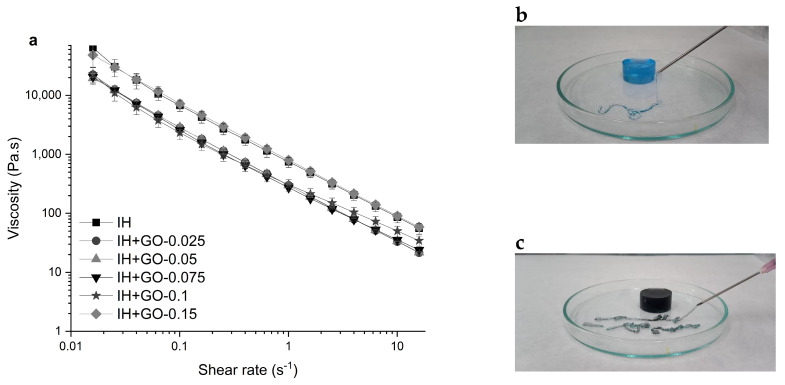
Apparent viscosity as a function of shear rate of (**a**) injectability test by hand of the hydrogels IH (**b**) and IH+GO-0.15 (**c**). The IH sample was stained with methylene blue to obtain an appropriate contrast in the image.

**Figure 5 polymers-16-02345-f005:**
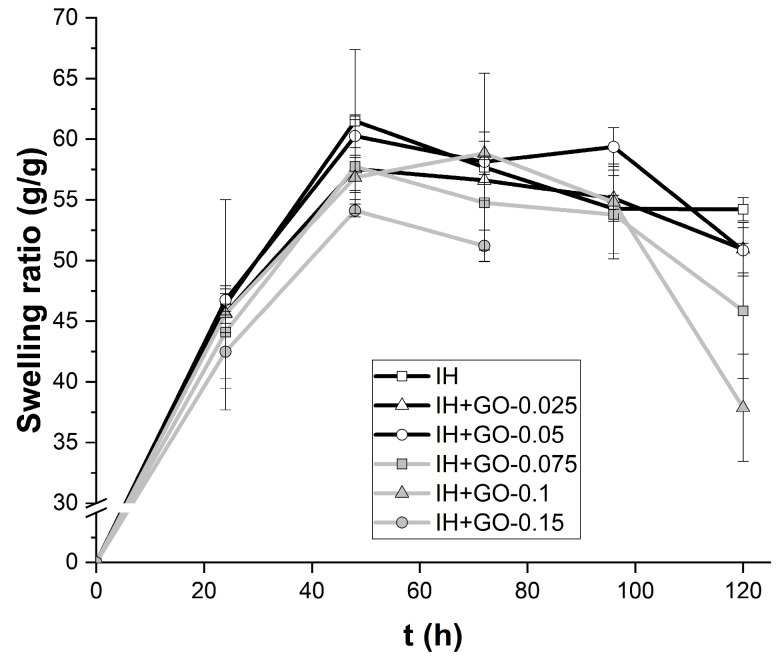
Effect of graphene oxide content on the swelling ratio of hydrogels.

**Figure 6 polymers-16-02345-f006:**
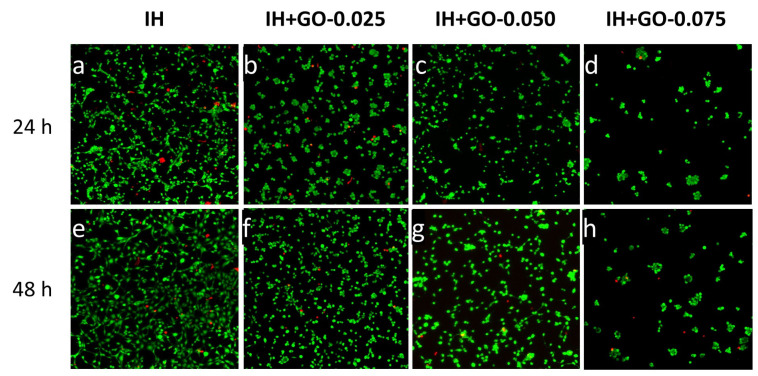
Confocal laser microscopy images obtained after 24 h (**a**–**d**) and 48 h (**e**–**h**) showing the live/dead staining of 3T3-L1 fibroblasts cultured with the hydrogels.

**Figure 7 polymers-16-02345-f007:**
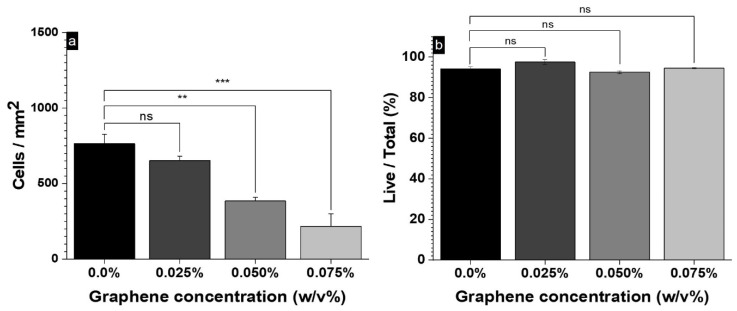
Count of live cells (**a**) and the live/total percentage (**b**) of the cells proliferated after 48 h inside the hydrogels with different graphene concentrations (ns = non-significant, ** *p* < 0.001, *** *p* < 0.0001).

**Table 1 polymers-16-02345-t001:** Concentration of graphene in each hydrogel.

Sample Code	GO *w*/*v* (%)
IH control	0
IH+GO-0.01	0.01
IH+GO-0.02	0.02
IH+GO-0.025	0.025
IH+GO-0.05	0.05
IH+GO-0.075	0.075
IH+GO-0.1	0.1
IH+GO-0.15	0.15

**Table 2 polymers-16-02345-t002:** G′/G″ ratio in each hydrogel.

Sample Code	G′/G″
IH	9
IH+GO-0.025	8
IH+GO-0.05	7
IH+GO-0.075	3
IH+GO-0.01	1
IH+GO-0.15	8

## Data Availability

Data are contained within the article and Appendix A.

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
