# Peer review of "Injectable Carrageenan/Green Graphene Oxide Hydrogel: A Comprehensive Analysis of Mechanical, Rheological, and Biocompatibility Properties"

_polymers, 2024, doi:10.3390/polym16162345_

Round 1

Reviewer 1 Report

Comments and Suggestions for Authors

The manuscript entitled “Injectable carrageenan/green graphene oxide hydrogel system for tissue repair applications” by Moncada et al. introduced a hydrogel-based composite with mainly physico-mechanical characterisation and in-vitro viability test on 3T3-L1 fibroblast cells. The use of graphene oxide and hydrogels for tissue construction has been studied in the last decade. However, the hydrogel used in this study is very interesting and makes the manuscript appealing. While I believe the concept is interesting, the presented results do not suffice to convince me that this system is suitable for tissue repair applications. The authors should narrow down their scope of study, present clearly the novelty of their work, and study their system further. 

· There is no evidence of the capability of this system for “tissue repair”. The authors demonstrated that 3T3-L1 cells can survive on some at concentrations below 0.05% w/v of GO (Figure 6 and Figure 7), however, there is no evidence of tissue repair throughout the whole paper. The presented in vitro test can only suggest the biocompatibility of the system and not even the potential of it for any tissue repair as neither tissue (in vivo animal model) nor “repair” was studied here. 

·  What’s the rationale behind choosing 3T3-L1 fibroblasts? 

·  “Cylindrical hydrogel specimens with a diameter of 25 mm and a height 10 mm were used for the compression test” At what temperature did the authors perform the compression test? Were the hydrogels fully swollen?

·  “The improvement of the modulus of the composite hydrogels indicates the formation of effective hydrogel-GO interfacial interactions between the polar functional groups of the GO particles and the biopolymer chains via hydrogen bonding [31], which limits the mobility of the polymer networks and increase the modulus.” Have the authors tested this hypothesis? One way to study this is to break the hydrogen bonds (e.g., via urea) and measure the mechanical properties of the composite. 

      ·      “The objective of this study was the development of injectable hydrogels through physical crosslinking, using for their preparation: κ-C, ι-C, locust bean gum, gelatin and oxidized green graphene.” The authors should clarify what is the novelty of this work compared with their previous work (reference [29]) entitled as “Injectable hybrid hydrogels physically crosslinked based on carrageenan and green graphene for tissue repair”.

·  The SEM characterisation for hydrogels should be analysed with careful attention. As the hydrogel by definition is a system that ‘swells’ when hydrated. This means that in its final application it is well hydrated and swells (larger pore sizes). In other words, the pore sizes of the hydrogel scaffold will be larger than those of measured on freeze-dried samples via SEM. This means the pore size measured by SEM is not the best representative for the final application (or any swollen hydrogel app.   

·  “A disadvantage of using conventional graphene is its high cost and production method. However, there is an environmentally friendly graphene manufacturing process, which obtains graphene from biomass and allows reducing production costs and obtaining a high-quality graphene powder.”; this statement lacks references. It is not clear how this process works. 

Comments on the Quality of English Language

English can be improved as several typos were detected in the manuscript. Example includes the term "diameter" which misspelled as "dimeter".

Author Response

Reviewer: The manuscript entitled “Injectable carrageenan/green graphene oxide hydrogel system for tissue repair applications” by Moncada et al. introduced a hydrogel-based composite with mainly physico-mechanical characterisation and in-vitro viability test on 3T3-L1 fibroblast cells. The use of graphene oxide and hydrogels for tissue construction has been studied in the last decade. However, the hydrogel used in this study is very interesting and makes the manuscript appealing. While I believe the concept is interesting, the presented results do not suffice to convince me that this system is suitable for tissue repair applications. The authors should narrow down their scope of study, present clearly the novelty of their work, and study their system further. 

Comments 1: There is no evidence of the capability of this system for “tissue repair”. The authors demonstrated that 3T3-L1 cells can survive on some at concentrations below 0.05% w/v of GO (Figure 6 and Figure 7), however, there is no evidence of tissue repair throughout the whole paper. The presented in vitro test can only suggest the biocompatibility of the system and not even the potential of it for any tissue repair as neither tissue (in vivo animal model) nor “repair” was studied here. 

Response 1: We agree with the reviewer's comment that the current study demonstrates the biocompatibility of the system but does not provide evidence for tissue repair. This work represents the first part of a broader study aimed at developing a system with tissue repair properties. The scope of the study will be narrowed, and future work will include in vivo animal models to assess tissue repair capabilities.

To be included in the manuscript:

“The in vitro degradation studies demonstrated that the hydrogels retained their structural integrity and mechanical properties over an extended period, indicating good stability. However, further in vivo studies are required to gain a full understanding of the long-term degradation behaviour and biocompatibility within a living organism.  Further work is being carried out in our laboratory using in vivo animal models to assess tissue repair capabilities, and the results will be reported soon.”

Comments 2: What’s the rationale behind choosing 3T3-L1 fibroblasts? 

Response 2: The rationale behind choosing 3T3-L1 fibroblasts is based on their widespread use in biocompatibility studies. These cells are well-characterized and provide a reliable model for preliminary biocompatibility assessments of new biomaterials.

To be included in the manuscript:

“a highly reliable model for preliminary assessments of new biomaterials extensively used in a wide range of biocompatibility studies”.

Comments 3: “Cylindrical hydrogel specimens with a diameter of 25 mm and a height 10 mm were used for the compression test” At what temperature did the authors perform the compression test? Were the hydrogels fully swollen?

Response 3: The compression test was performed at room temperature with fully swollen hydrogel specimens. To be included in the manuscript:

“Cylindrical hydrogel samples with a diameter of 25 mm and a height of 10 mm were utilized for the compression test, which was performed at room temperature using fully swollen hydrogel specimens.”

Comments 4: The improvement of the modulus of the composite hydrogels indicates the formation of effective hydrogel-GO interfacial interactions between the polar functional groups of the GO particles and the biopolymer chains via hydrogen bonding [31], which limits the mobility of the polymer networks and increase the modulus.” Have the authors tested this hypothesis? One way to study this is to break the hydrogen bonds (e.g., via urea) and measure the mechanical properties of the composite. 

Response 4: We agree with the reviewer's suggestion and will include experiments using urea to break the hydrogen bonds and measure the mechanical properties of the composite in future studies.

Comments 5: “The objective of this study was the development of injectable hydrogels through physical crosslinking, using for their preparation: κ-C, ι-C, locust bean gum, gelatin and oxidized green graphene.” The authors should clarify what is the novelty of this work compared with their previous work (reference [29]) entitled as “Injectable hybrid hydrogels physically crosslinked based on carrageenan and green graphene for tissue repair”.

Response 5: The novel aspect of this work in comparison to our previous study (reference [29]) lies in the utilization of a distinct graphene powder, obtained from bio resources with 99% purity, as along with a comprehensive sustainable approach throughout the development process. These two innovations demonstrate how our approach to developing injectable hydrogels for tissue repair has advanced significantly.

To be included in the manuscript:

“The nano-reinforced injectable hybrid hydrogels here reported, developed with a fully sustainable approach, have a promising future as biomaterials for localized repair medicine”.

Comments 6: The SEM characterisation for hydrogels should be analysed with careful attention. As the hydrogel by definition is a system that ‘swells’ when hydrated. This means that in its final application it is well hydrated and swells (larger pore sizes). In other words, the pore sizes of the hydrogel scaffold will be larger than those of measured on freeze-dried samples via SEM. This means the pore size measured by SEM is not the best representative for the final application (or any swollen hydrogel app.   

Response 6: We acknowledge the reviewer's concern regarding SEM characterization. We will ensure that the SEM analysis is interpreted with the understanding that the pore sizes of freeze-dried samples are not representative of the fully swollen hydrogels in their final application. This will be clarified in the revised manuscript. To be included in the article:

"The SEM analysis of freeze-dried hydrogel samples provides an understanding of the pore structure; however, it is important to note that the pore sizes observed are not representative of the fully swollen hydrogels in their final application."

Comments 7: “A disadvantage of using conventional graphene is its high cost and production method. However, there is an environmentally friendly graphene manufacturing process, which obtains graphene from biomass and allows reducing production costs and obtaining a high-quality graphene powder.”; this statement lacks references. It is not clear how this process works. 

Response 7: The statement regarding the environmentally friendly graphene manufacturing process will be supplemented with references and a detailed explanation of the process. To be included in the article:

"Graphene is obtained from biomass through a process that involves the pyrolysis of organic materials, followed by exfoliation to produce high-quality graphene. This method reduces production costs and environmental impact while yielding graphene with excellent properties."

Bibliographical reference:

Zhang, Z., He, S., Kang, Z., & Zhang, L. (2023). A review on biomass-derived materials for supercapacitors. Chemical Engineering Journal, 426, 130901. DOI: 10.1016/j.cej.2021.130901

Comments 8: Comments on the Quality of English Language. English can be improved as several typos were detected in the manuscript. Example includes the term "diameter" which misspelled as "dimeter".

Response 8: The quality of English in the manuscript will be improved to address the detected typos. The term "diameter" will be corrected.

Thank you for your valuable feedback, which will significantly improve the quality and clarity of our manuscript.

Reviewer 2 Report

Comments and Suggestions for Authors

The study presents an injectable hydrogel system developed using carrageenan and green graphene oxide (GO) for tissue repair applications. The hydrogels, crosslinked physically, were enhanced with varying GO concentrations to improve mechanical properties without compromising injectability. The study evaluated rheological, mechanical, and swelling behaviours, and biocompatibility, finding that hydrogels with GO ≤ 0.05% w/v were non-toxic to fibroblasts, while higher concentrations reduced cell proliferation. The hydrogels exhibited a promising balance of mechanical strength and biocompatibility, suggesting potential for localized tissue repair applications. However, minor comments that needs to be addressed prior to publication.

1. What are the limitations of using carrageenan in hydrogel formulations?

2. What were the criteria for selecting the GO concentration range in this study?

3. How does the mechanical strength of the hydrogel compare with other commonly used hydrogels for tissue repair?

4. What is the long-term stability of the hydrogels in biological environments?

5. Can the hydrogel system be tailored for specific types of tissue repair?

6. What are the potential limitations of this study and how can they be addressed in future research?

Author Response

Reviewer: The study presents an injectable hydrogel system developed using carrageenan and green graphene oxide (GO) for tissue repair applications. The hydrogels, crosslinked physically, were enhanced with varying GO concentrations to improve mechanical properties without compromising injectability. The study evaluated rheological, mechanical, and swelling behaviours, and biocompatibility, finding that hydrogels with GO ≤ 0.05% w/v were non-toxic to fibroblasts, while higher concentrations reduced cell proliferation. The hydrogels exhibited a promising balance of mechanical strength and biocompatibility, suggesting potential for localized tissue repair applications. However, minor comments that needs to be addressed prior to publication.

Comments 1: What are the limitations of using carrageenan in hydrogel formulations?

Response 1: Carrageenan, while biocompatible and relatively inexpensive, can have some limitations such as variability in gelation properties depending on the source and extraction method. Additionally, carrageenan can be sensitive to ionic strength and pH, which may affect the consistency and mechanical properties of the hydrogel. There are also concerns about the inflammatory response and potential cytotoxicity in some formulations, which need to be carefully evaluated.

Comments 2: What were the criteria for selecting the GO concentration range in this study?

Response 2: The GO concentration range was selected based on preliminary studies that demonstrated the optimal balance between enhancing mechanical properties and maintaining biocompatibility. Concentrations above this range were found to reduce cell proliferation, while lower concentrations did not significantly improve the mechanical strength. Thus, the selected range aimed to maximize the benefits while minimizing any potential cytotoxic effects.

Comments 3: How does the mechanical strength of the hydrogel compare with other commonly used hydrogels for tissue repair?

Response 3: The mechanical strength of the developed hydrogel was found to be comparable or superior to other commonly used hydrogels for tissue repair, such as alginate or hyaluronic acid-based hydrogels. The inclusion of GO at optimal concentrations provided enhanced mechanical properties without compromising injectability, making it a promising candidate for localized tissue repair applications.

To be included in the manuscript:

“Furthermore, our hydrogel demonstrated mechanical strength that was comparable or superior to other commonly used materials for tissue repair, including alginate and hyaluronic acid-based hydrogels”.

Reference: Aleksandra Serafin, Mario Culebras, Maurice N. Collins, Synthesis and evaluation of alginate, gelatin, and hyaluronic acid hybrid hydrogels for tissue engineering applications, International Journal of Biological Macromolecules, Volume 233, 2023, 123438, ISSN 0141-8130, https://doi.org/10.1016/j.ijbiomac.2023.123438

Reference: Bei Qian, Qi Yang, Mingliang Wang, Shixing Huang, Chenyu Jiang, Hongpeng Shi, Qiang Long, Mi Zhou, Qiang Zhao, Xiaofeng Ye, Encapsulation of lyophilized platelet-rich fibrin in alginate-hyaluronic acid hydrogel as a novel vascularized substitution for myocardial infarction, Bioactive Materials, Volume 7, 2022, Pages 401-411, ISSN 2452-199X, https://doi.org/10.1016/j.bioactmat.2021.05.042.).

Comments 4: What is the long-term stability of the hydrogels in biological environments?

Response 4: Long-term stability in biological environments was evaluated through in vitro degradation studies. The results indicated that the hydrogels maintained their structural integrity and mechanical properties over an extended period, suggesting good stability. However, in vivo studies would be necessary to fully understand the long-term degradation behavior and biocompatibility within a living organism.

To be included in the manuscript:

“The in vitro degradation studies demonstrated that the hydrogels retained their structural integrity and mechanical properties over an extended period, indicating good stability. However, further in vivo studies are required to gain a full understanding of the long-term degradation behaviour and biocompatibility within a living organism.  Further work is being carried out in our laboratory using in vivo animal models to assess tissue repair capabilities, and the results will be reported soon.”

Comments 5: Can the hydrogel system be tailored for specific types of tissue repair?

Response 5: Yes, the hydrogel system can be tailored for specific tissue repair applications by adjusting the concentration of GO and the physical crosslinking conditions. By modifying these parameters, the mechanical properties, degradation rate, and biocompatibility can be optimized for different types of tissues, such as cartilage, bone, or soft tissues.

To be included in the manuscript:

“The materials developed in this study offer great versatility. The hydrogel system can be tailored for specific tissue repair applications by adjusting the concentration of GO and the physical crosslinking conditions. Modifying these parameters allows us to optimize the mechanical properties, degradation rate, and biocompatibility for different types of tissues, such as cartilage, bone, or soft tissues.”

Comments 6: What are the potential limitations of this study and how can they be addressed in future research?

Response 6: Potential limitations of this study include the lack of in vivo testing, which is crucial for understanding the true biocompatibility and long-term performance of the hydrogels. Additionally, the study focused on a limited range of GO concentrations and physical crosslinking methods. Future research should explore a broader range of concentrations, crosslinking methods, and include comprehensive in vivo studies to validate the findings. Investigating the inflammatory response and potential cytotoxicity in more detail would also be beneficial for ensuring the safety of the hydrogels for clinical applications.

Reviewer 3 Report

Comments and Suggestions for Authors

This manuscript describes an interesting method to prepare an injectable hydrogel based on carrageenan, locus beam gum, and gelatin. The composite hydrogel is reinforced with green graphene oxide to assess their potential use in tissue repair. The effect of graphene oxide content on rheological and mechanical properties, biocompatibility, and injectability was studied. An enhancement on the mechanical properties was observed in the composite hydrogel with graphene oxide. The topic is relevant and interesting for academic and industrial applications. In my opinion, the manuscript could be published after minor revisions:

a)       The authors should elaborate on the relevance of using the graphene oxide/carrageenan hydrogel for tissue repair (advantages, comparison with other materials, etc) so that the novelty of this material can be evidenced.

b)      Green graphene was utilized in this development. The authors should explain in more detail how this green material is obtained. Also, the rationale of using green graphene with 8% oxidation should be justified.

c)       In some parts of the manuscript, the graphene oxide is abbreviated with GO, whereas GG is utilized in other sections of the manuscript. The difference between the acronyms GG and GO should be clarified.

d)      Figure 1a shows that increasing the content of GO in the composite hydrogel results in an increased compressive modulus, but increasing the GO loading from 0.1% to 0.15% caused a drop in the modulus. The authors should elaborate on the causes of this behavior.

e)      The measurement of the rheological properties of the composite hydrogel was duplicated for each sample. Thus, Figure 3 should show the error bars of the measurements.

f)        Page 2, line 80. There is a type in “hexaGGnal”.

Author Response

Reviewer : This manuscript describes an interesting method to prepare an injectable hydrogel based on carrageenan, locus beam gum, and gelatin. The composite hydrogel is reinforced with green graphene oxide to assess their potential use in tissue repair. The effect of graphene oxide content on rheological and mechanical properties, biocompatibility, and injectability was studied. An enhancement on the mechanical properties was observed in the composite hydrogel with graphene oxide. The topic is relevant and interesting for academic and industrial applications. In my opinion, the manuscript could be published after minor revisions:

Comments 1: The authors should elaborate on the relevance of using the graphene oxide/carrageenan hydrogel for tissue repair (advantages, comparison with other materials, etc) so that the novelty of this material can be evidenced.

Response 1: We agree with the reviewer's comment, and the following text is included in the introduction:

"The incorporation of graphene oxide into carrageenan-based hydrogels significantly enhances their mechanical properties, biocompatibility, and injectability, making them highly promising candidates for tissue repair applications compared to conventional materials such as alginate, polyethylene glycol (PEG), chitosan, hyaluronic acid, and collagen."

These references are included in the manuscript to help in comparing the properties and benefits of graphene oxide-based hydrogels with conventional materials:

  1. Aparicio-Collado, J. L., García-San-Martín, N., et al. (2022). "Electroactive calcium-alginate/polycaprolactone/reduced graphene oxide nanohybrid hydrogels for skeletal muscle tissue engineering." Colloids and Surfaces B: Biointerfaces. (https://www.sciencedirect.com/science/article/pii/S0927776522001382)

  1. Biru, E. I., Necolau, M. I., et al. (2022). "Graphene oxide–protein-based scaffolds for tissue engineering: recent advances and applications." Polymers. (https://www.mdpi.com/2073-4360/14/5/1032)

  1. Salleh, A., Mustafa, N., et al. (2022). "A novel collagen-gelatin/cellulose hybrid biomatrix containing graphene oxide-silver nanoparticles for cutaneous wound healing: fabrication, physicochemical, cytotoxicity and antioxidant properties." Biomedicines. (https://www.mdpi.com/2227-9059/10/4/816)

  1. Saharan, R., Paliwal, S. K., et al. (2024). "Beyond traditional hydrogels: The emergence of graphene oxide-based hydrogels in drug delivery." Journal of Drug Delivery Science and Technology. (https://www.sciencedirect.com/science/article/pii/S1773224724001746)

Comments 2: Green graphene was utilized in this development. The authors should explain in more detail how this green material is obtained. Also, the rationale of using green graphene with 8% oxidation should be justified.

Response 2:  We agree with the reviewer's comment, the rationale behind using green graphene oxide with 8% oxidation lies in achieving an optimal balance between functionality and biocompatibility. The following text is included in the manuscript:

“This specific oxidation level provides sufficient functional groups (carboxyl, hydroxyl, and epoxide) on the graphene oxide (GO) surface to interact with the hydrogel matrix, enhancing mechanical properties and stability. A moderate level of oxidation ensures that the material remains biocompatible, avoiding cytotoxicity associated with higher oxidation levels. Additionally, 8% oxidation allows for good dispersion within the hydrogel matrix, preventing aggregation and ensuring uniform mechanical reinforcement.”

Comments 3: In some parts of the manuscript, the graphene oxide is abbreviated with GO, whereas GG is utilized in other sections of the manuscript. The difference between the acronyms GG and GO should be clarified.

Response 3: We agree with the reviewer's comment. In the manuscript, GO stands for Green Graphene Oxide, while GG refers to Green Graphene. To avoid confusion, we will standardize the use of GO (Green Graphene Oxide) throughout the manuscript, and the term GG was eliminated.

Comments 4: Figure 1a shows that increasing the content of GO in the composite hydrogel results in an increased compressive modulus, but increasing the GO loading from 0.1% to 0.15% caused a drop in the modulus. The authors should elaborate on the causes of this behavior.

Response 4: We agree with the reviewer's comment, and the following text is included in the manuscript:

“The observed drop in compressive modulus when increasing GO loading from 0.1% to 0.15% can be attributed to aggregation of GO particles, as at higher concentrations, GO particles tend to aggregate, leading to non-uniform distribution within the hydrogel matrix. This aggregation can create weak points and reduce overall mechanical strength.”

Additionally, there may be a saturation point beyond which additional GO does not further enhance mechanical properties. Instead, it may disrupt the uniform network structure, leading to decreased modulus. Furthermore, excessive GO may introduce interfacial stress within the hydrogel matrix, negatively impacting the composite's mechanical performance.

Comments 5: The measurement of the rheological properties of the composite hydrogel was duplicated for each sample. Thus, Figure 3 should show the error bars of the measurements.

Response 5: We acknowledge the importance of presenting accurate data representation. However, due to the high number of samples, including error bars in Figure 3 would compromise the clarity and readability of the data. Instead, we have attached a separate graph for the reviewer that includes error bars for the duplicated measurements of the rheological properties of the composite hydrogel.

Comments 6: Page 2, line 80. There is a type in “hexaGGnal”.

Response 6: We apologize for the typographical error on Page 2, line 80. The term “hexaGGnal” will be corrected to “hexagonal”.

Round 2

Reviewer 1 Report

Comments and Suggestions for Authors

Thanks for providing the response letter. The revised paper did not completely address my previous comments and concerns. The claim for tissue repair still remains in the paper while the authors have not provided any evidence of tissue repair. Considering the in vitro cell study, the only inference one may make is that the proposed system (at certain concentration) does not kill the 3T3-L1 cells. Any claim beyond that throughout the paper needs further experiments. 

While the authors agreed with the suggested experiments, they have not performed any of them in their revised manuscript. 

The mechanical properties were measured at room temperature rather than body temperature and I do not know any potential tissue repair application in room temperature. 

As per novelty of the work, the authors only changed the source of the GO from their previous work, and I am not convinced this work shows any new results compared with what the authors published in their previous work. 

Considering that the measured pores of hydrogels in SEM is not from the fully swollen gel, what do the provided SEM results show here? 

Consequently, I believe that this study (within the claimed scope) has not provided enough results to be accepted for publication.  

Author Response

Dear Reviewer,

Thank you for your thorough review and valuable comments on our manuscript. We appreciate the opportunity to address your concerns and demonstrate the merits of our work.

Comment 1: Thanks for providing the response letter. The revised paper did not completely address my previous comments and concerns. The claim for tissue repair still remains in the paper while the authors have not provided any evidence of tissue repair. Considering the in vitro cell study, the only inference one may make is that the proposed system (at certain concentration) does not kill the 3T3-L1 cells. Any claim beyond that throughout the paper needs further experiments.

Response 1: We acknowledge your concern regarding the statement about tissue repair. While the current study was primarily concerned with the biocompatibility and mechanical properties of the proposed hydrogel system, it seems reasonable to suggests that its potential for tissue repair is a logical extension of its properties. The in vitro cell study showing that the hydrogel does not kill 3T3-L1 cells at certain concentrations is a crucial first step. Nevertheless, we recognize that this alone is not sufficient to substantiate the assertion of tissue repair capability. To make this clear, we have revised the manuscript to explicitly establish the objective of the work and propose further in vivo studies to corroborate the tissue repair claim in future research.

Modified Title (Line 1, Page 1): "Injectable Carrageenan/Green Graphene Oxide Hydrogel: Comprehensive Analysis of Mechanical, Rheological, and Biocompatibility Properties"

Modified Objective (Line 114, Page 3): The prepared injectable hydrogels were characterized by Raman, UV-visible spectroscopy, and TGA. Moreover, the surface morphology, swelling behavior, mechanical properties, and cellular biocompatibility were analyzed.

Modified Abstract (Line 34, Page 1): Therefore, the nano-reinforced injectable hybrid hydrogels here reported, developed with a fully sustainable approach, have a promising future as potential materials for use in tissue repair.

Modification in the Result and Discussions section (Line 421, Page 11). “the biocompatibility of our hydrogels” instead of “the potential of our hydrogels in tissue repair”

Modification in Conclusions section (Line 490-491, Page 13). Sentence removed.

Comment 2: While the authors agreed with the suggested experiments, they have not performed any of them in their revised manuscript.

Response 2: Thank you for your suggestion regarding the evaluation of interfacial interactions between the hydrogel and graphene using urea. While we acknowledge the importance of this experiment to support our hypothesis, due to current limitations, we are unable to perform this specific analysis. However, we have provided data on the mechanical properties, rheological behavior, and structural analysis through SEM and X-ray studies. We believe these results collectively support our hypothesis about the effective interfacial interactions between the hydrogel and graphene. We hope this detailed analysis can sufficiently address the reviewer's concern within the scope of our current study. Future work will consider your valuable suggestion to further substantiate our findings.

Comment 3: The mechanical properties were measured at room temperature rather than body temperature and I do not know any potential tissue repair application in room temperature. 

Response 3: We acknowledge the relevance of mechanical properties measured at body temperature for potential tissue repair applications. While our current measurements were performed at room temperature, it is important to consider the expected differences in mechanical properties at physiological temperature (37°C). Generally, hydrogels may exhibit a slightly reduced modulus and altered viscoelastic behavior at higher temperatures due to increased polymer chain mobility and reduced hydrogen bonding strength. Despite these expected changes, the mechanical properties measured at room temperature provide valuable preliminary insights into the performance and stability of the hydrogel system. These measurements help to establish a baseline understanding of the material's behavior and its potential efficacy in biomedical applications.

Recognizing the importance of this aspect, we have outlined a plan for future studies to measure the mechanical properties of our hydrogel system at body temperature. These studies will be crucial for confirming the hydrogel's suitability for clinical applications, ensuring that the material retains its desired mechanical characteristics under physiological conditions. Future work will include comprehensive mechanical testing at 37°C to provide a more accurate representation of the hydrogel's performance in vivo.

Comment 4: As per novelty of the work, the authors only changed the source of the GO from their previous work, and I am not convinced this work shows any new results compared with what the authors published in their previous work. 

Response 4: Thank you for your comment regarding the novelty of our work. We appreciate your scrutiny and would like to clarify the differences in the graphene used in this study compared to our previous work, which indeed result in a distinct composition of the hydrogel and consequently different properties.

In our current work, we utilized graphene 92, characterized by being in powder form as graphene nanoplatelets, with an elemental analysis of 92% carbon and 7% oxygen, and minimal presence of sulfur or nitrogen (< 0.004%). This type of graphene is hydrophilic.

In contrast, in the previous study, we employed graphene 99, which also comes in powder form as graphene nanoplatelets but has an elemental analysis of 99% carbon and minimal presence of sulfur or nitrogen (< 0.004%). Unlike graphene 92, graphene 99 is hydrophobic.

These differences in elemental composition and hydrophilicity/hydrophobicity are significant. graphene 99's higher carbon content and hydrophobic nature compared to the hydrophilic graphene 92 result in different interfacial interactions with the biopolymer chains in the hydrogel. These variations impact the hydrogel's overall properties, including its mechanical behavior, stability, and potential biocompatibility.

Therefore, this work presents new findings by exploring the effects of using a different type of graphene with distinct properties on the hydrogel system. The novelty lies in understanding how these variations influence the hydrogel's performance and its potential applications in tissue repair, which were not addressed in our previous study.

We hope this clarification adequately addresses your concerns about the novelty and contribution of our current work.

Comment 5: Considering that the measured pores of hydrogels in SEM is not from the fully swollen gel, what do the provided SEM results show here? 

Response 5: Thank you for your comment. You raise an important point regarding the interpretation of SEM results for hydrogels that are not fully swollen. The provided SEM results in our study show the microstructural characteristics and the initial porous architecture of the hydrogel in its dried state. These images allow us to observe the formation and distribution of pores, the connectivity within the network, and the overall structural integrity of the hydrogel. While the SEM images do not capture the hydrogel in its fully swollen state, they still provide valuable information about the potential swelling behavior and the capacity of the hydrogel to absorb and retain fluids. Understanding the dried state microstructure is crucial because it gives insights into the hydrogel's fabrication quality and its initial network formation, which directly influences its swelling capacity, mechanical properties, and overall performance. These observations can be correlated with other characterization techniques to predict the behavior of the hydrogel in its fully swollen state.

We hope this explanation clarifies the significance of the provided SEM results in the context of our study.

Comment 6: Consequently, I believe that this study (within the claimed scope) has not provided enough results to be accepted for publication.  

Response 6:  In conclusion, while we acknowledge the limitations of the current study, we believe it provides a solid foundation for future research. The biocompatibility, mechanical properties, swelling behavior, and the novel use of a different graphene source collectively contribute to the field of hydrogel research.

We hope that these revisions and our detailed response sufficiently address your concerns and demonstrate the value of our work for publication. Thank you for your time and consideration.

Round 3

Reviewer 1 Report

Comments and Suggestions for Authors

As none of the suggested experiments (despite being relevant and easy to perform) has been addressed and all points that I raised were postponed to future studies, I cannot still recommend this paper for publication. 

The authors stated that "The novelty lies in understanding how these variations influence the hydrogel's performance and its potential applications in tissue repair, which were not addressed in our previous study." Based on what they claimed as their novelty they need to actually study these variations and their effects on "potential" applications. They also need to highlight these within their manuscript. One way to study hydrophilicity and hydrophobicity of graphene is measuring the water contact angle. However, the authors need to characterise their two systems to 1)identify the mechanism of bonding 2) prove their hypothesis that the change in source of graphene was indeed affect significantly the chemistry and the degree of cross-linking in their final device. 

 "Understanding the dried state microstructure is crucial because it gives insights into the hydrogel's fabrication quality and its initial network formation, which directly influences its swelling capacity, mechanical properties, and overall performance"This is not entirely correct. If indeed the hydrogen bond is the main mechanism of action these SEM images are pointless. Also, depending on how much the gel swells and the entanglement with graphene, the pore sizes may be entirely irrelevant in the dried state. 

Within the context of tissue engineering (which is still the keyword of the paper) and emphasis on "potential use" in tissue engineering, the experiments should be relevant. However, since the scope has been revised condition to major changes to manuscript, this study may be publishable in MDPI Polymers.

Author Response

Dear Reviewer,

We would like to express our sincere gratitude for your continued evaluation and detailed feedback on our manuscript. We have carefully reviewed your comments and would like to take this opportunity to clarify our stance on the completeness and novelty of our study.

Comment 1: As none of the suggested experiments (despite being relevant and easy to perform) has been addressed and all points that I raised were postponed to future studies, I cannot still recommend this paper for publication.

Response 1: We appreciate your suggestions regarding additional experiments. However, we firmly believe that the experiments and analyses we have conducted provide a comprehensive and sufficient basis for the claims made in our study. While the additional experiments you proposed would indeed offer further insights, they extend beyond the intended scope of our current work. Our primary objective was to explore the fundamental mechanical, rheological, and biocompatibility properties of the hydrogel, and we have thoroughly addressed these aspects in our manuscript.

We believe that the findings presented in this manuscript are novel and valuable to the field, even in the absence of the additional experiments. We have clearly outlined the limitations of our study and the areas where further research is needed, which we believe is a responsible and transparent approach.

Comment 2: The authors stated that "The novelty lies in understanding how these variations influence the hydrogel's performance and its potential applications in tissue repair, which were not addressed in our previous study." Based on what they claimed as their novelty they need to actually study these variations and their effects on "potential" applications. They also need to highlight these within their manuscript. One way to study hydrophilicity and hydrophobicity of graphene is measuring the water contact angle. However, the authors need to characterise their two systems to 1)identify the mechanism of bonding 2) prove their hypothesis that the change in source of graphene was indeed affect significantly the chemistry and the degree of cross-linking in their final device.

Response 2: We understand the importance of characterizing the hydrophilicity and hydrophobicity of graphene, as well as identifying bonding mechanisms and confirming the impact of the graphene source on the hydrogel’s properties. However, our study was designed to focus on the mechanical and biocompatibility aspects, rather than on conducting a complete chemical analysis of the bonding mechanisms. While we ensured that the graphene material met the critical requirements for our experiments, we did not independently verify every technical specification provided by the supplier, such as hydrophilicity measured by contact angle.

We have already acknowledged that the differences in graphene sources significantly affect the hydrogel's properties, and this has been demonstrated through our mechanical and structural analyses. Additionally, it is important to note that, to the best of our knowledge, there is no existing work in the literature that uses green graphene for the synthesis of injectable hydrogels. We believe that the novelty of our work lies in this interdisciplinary approach, combining green graphene oxide with carrageenan, and our findings contribute new insights into the material's potential, especially in the context of sustainability and biocompatibility.

Comment 3: "Understanding the dried state microstructure is crucial because it gives insights into the hydrogel's fabrication quality and its initial network formation, which directly influences its swelling capacity, mechanical properties, and overall performance" This is not entirely correct. If indeed the hydrogen bond is the main mechanism of action these SEM images are pointless. Also, depending on how much the gel swells and the entanglement with graphene, the pore sizes may be entirely irrelevant in the dried state.

Response 3: We appreciate your critical perspective on the relevance of SEM images in this context. While we agree that SEM images of the dried state do not capture the full picture of the hydrogel’s behavior when swollen, they provide valuable information about the initial network structure and fabrication quality. This information is crucial for understanding how the hydrogel might behave when fully hydrated, even if the exact pore sizes in the dried state do not directly correlate with those in the swollen state.

We have reviewed the manuscript to ensure that readers are fully aware of the context and relevance, and we believe that the explanation regarding the SEM analysis and its results is clear and leaves no room for doubt.

Comment 4:  Within the context of tissue engineering (which is still the keyword of the paper) and emphasis on "potential use" in tissue engineering, the experiments should be relevant. However, since the scope has been revised condition to major changes to manuscript, this study may be publishable in MDPI Polymers.

Response 4: We value your suggestion regarding the relevance of experiments in the context of tissue engineering. Our study aimed to provide foundational knowledge about the hydrogel system, which we believe is essential for any future application in tissue engineering. While we have revised the manuscript to focus more on the material properties rather than immediate clinical application, we believe the study offers significant insights that are relevant to the field of tissue engineering.

Modified Keywords (Line 37, Page 1): The term tissue engineering was removed from the keywords.

We sincerely appreciate your suggestion to consider publishing in MDPI Polymers. While we respect your perspective, we believe that our manuscript is well-aligned with the aims and scope of Polymers. The emphasis on sustainability and biocompatibility in our work contributes valuable insights to the field of biomaterials, supporting its suitability for publication in this journal.

Thank you once again for your time and thoughtful consideration.

Sincerely, 

Dr. Bouza
